# Molecular Dynamics Simulation of Transport Mechanism of Graphene Quantum Dots through Different Cell Membranes

**DOI:** 10.3390/membranes12080753

**Published:** 2022-07-31

**Authors:** Pengzhen Zhang, Fangfang Jiao, Lingxiao Wu, Zhe Kong, Wei Hu, Lijun Liang, Yongjun Zhang

**Affiliations:** 1Center of Advanced Optoelectronic Materials and Devices, Key Laboratory of Novel Materials for Sensor of Zhejiang Province, College of Materials and Environmental Engineering, Hangzhou Dianzi University, Hangzhou 310018, China; 13519288162@163.com (P.Z.); wlx_1171999418@163.com (L.W.); yjzhang@hdu.edu.cn (Y.Z.); 2Shandong Provincial Key Laboratory of Molecular Engineering, School of Chemistry and Pharmaceutical Engineering, Qilu University of Technology (Shandong Academy of Sciences), Jinan 250353, China; jiao1313@outlook.com; 3College of Automation, Hangzhou Dianzi University, Hangzhou 310018, China; michael.lijunl@gmail.com

**Keywords:** graphene quantum dots, lipid membrane, molecular dynamics simulation, phospholipid

## Abstract

Exploring the mechanisms underlying the permeation of graphene quantum dots (GQDs) through different cell membranes is key for the practical application of GQDs in medicine. Here, the permeation process of GQDs through different lipid membranes was evaluated using molecular dynamics (MD) simulations. Our results showed that GQDs can easily permeate into 1-palmitoyl-2-oleoyl-sn-glycero-3-phosphocholine (POPC) and 1,2-dioleoyl-sn-glycero-3-phosphoethanolamine (DOPE) lipid membranes with low phospholipid molecule densities but cannot permeate into 1-palmitoyl-2-oleoyl phosphatidylethanolamine (POPE) lipid membranes with high phospholipid densities. Free energy calculation showed that a high-energy barrier exists on the surface of the POPE lipid membrane, which prevents GQDs from entering the cell membrane interior. Further analysis of the POPE membrane structure showed that sparsely arranged phospholipid molecules of the low-density lipid membrane facilitated the entry of GQDs into the interior of the membrane, compared to compactly arranged molecules in the high-density lipid membrane. Our simulation study provides new insights into the transmembrane transport of GQDs.

## 1. Introduction

As a new quasi-zero-dimensional nanomaterial, graphene quantum dots (GQDs) show broad applicability prospects in many fields owing to their ultra-small size, significant quantum size effects, and good biocompatibility [1,2,3,4,5,6]. GQDs are derived from graphene and graphene oxide (GO) and have excellent characteristics similar to graphene and GO, including mechanical and stable chemical properties [7,8,9,10]. In recent years, GQDs have been studied for application in the fields of biomedicine drug transmission and disease diagnosis [11,12,13]. Recently, GO has been used as an ideal drug-loading ligand to achieve targeted therapy, providing new prospects regarding the treatment of Parkinson’s disease [14]. Previous studies have shown that GQDs and related matrices can be used as platforms for cell adhesion, proliferation, and differentiation [15]. Similarly, GQDs and GO exhibit certain characteristics that make them suitable for anticancer therapy. For example, GO can be used to carry and effectively deliver doxorubicin to tumor cells [16].

The cytotoxicity of GQDs is lower than that of GO, so GQDs are suitable for biomedical applications [17]. Relevant studies showed that 3-nanometer GQDs had good membrane permeability, while 12-nanometer GQDs had weak permeability [18]. In addition, small-sized GQDs can be used for drug delivery because they can easily enter the intracellular space to attack tumor cells, eventually leading to tumor cell apoptosis [19]. Stable sulfur-doped GQDs can be used as efficient cell-imaging materials and can penetrate the cell membrane of HeLa cells [20]. As a crucial part of the cell, the cell membrane is not only a protective barrier to the cell but also plays an important role regarding compound exchange and metabolism. Some studies reported that GQDs have potential applications in medicine and can enter cells through direct infiltration, but the variation between organisms in terms of cell membrane structure should be considered [21,22]. The main lipid component of most bacteria is 1-palmitoyl-2-oleoyl phosphatidylethanolamine (POPE). For example, *Escherichia coli* (*E. coli*) contains 70–80% POPE, 20–25% POPG, and <5% cardiolipin [18,23]. In animal cells, the 1-palmitoyl-2-oleoyl-sn-glycero-3-phosphocholine (POPC) phospholipid is the main component of the cell membrane, and the POPE content in blood cells is 6% [24,25]. Similarly, the neutral 1,2-dioleoyl-sn-glycero-3-phosphoethanolamine (DOPE) cell membrane plays a significant role in gene transmission in biological cells [26]. However, the mechanisms underlying the interactions between GQDs and different cell membranes remain unclear. Therefore, understanding the permeation process of GQDs through different cell membranes is important for the advancement of GQD drug-loading therapy and biosensor research. 

In addition to experimental approaches, molecular dynamics (MD) simulations are frequently used to explore the mechanisms of interaction between nanomaterials and biological macromolecules. MD can realize the visualization of atoms and visually observe the details of the interaction between nanoparticles and biomolecules (including proteins, DNA, and cell membranes) [22,27,28,29]. For example, Ji et al. found that GQDs and GO can help drug molecules to enter cell membranes [30]. Moreover, a complex of GO and anticancer drug molecules can be transported to specific cancer cells [31]. Similarly, MD simulations were used previously to explore the interaction mechanism between GO and 5-fluorouracil (5-FU) [32]. In this study, MD simulations were used to explore the penetration mechanism of GQDs through different cell membranes, including POPC, DOPE, and POPE. See the Appendix A for simulation details and parameters of the system.

## 2. Computational Details

### 2.1. System Details

As described in our previous work [33,34,35], a graphene sheet was set on the *xy* plane, and a carbon atom was selected as the central atom (Cartesian coordinates 0, 0, 0). As shown in Figure 1, carbon atoms with x^2^ + y^2^ < R^2^ are part of the GQDs, where R is the radius of the GQDs [36]. In the MD simulation, the radius of the GQDs was set to 0.85 nm, the outermost carbon atoms of the GQDs were saturated with hydrogen atoms, and geometrical optimization based on density functional theory (DFT) was performed at the B3LYP/6-31G (d) level by Gaussian 03 [37]. Optimized GQDs were used for the initial MD configuration. Standardized lipid membranes were from CHARMM-GUI [38,39]. The varying areas of the different types of phospholipid molecules resulted in differences in the number of phospholipid molecules per unit area [25,40,41,42,43,44]. The detailed cell membrane information is presented in Table 1. Comparative studies of POPE and other phospholipids (POPC and DOPE) have shown that the area of POPE phospholipid molecules is the lowest [25]. The initialized GQDs were placed directly above the cell membrane with a distance of 4.5 nm from the centroid of the cell membrane. 

### 2.2. PMF Calculations

Based on the MD simulations, the free energy of GQDs’ translocation on different cell membranes was calculated using umbrella sampling. The zero coordinate was considered along the centroid of the cell membrane and 4.5 nm along the upper axis of the z-axis, and each equidistant window of 0.1 nm was extracted as the simulated reaction configuration. A harmonic force constant of 1000 kJmol^−1^nm ^−2^ was used to position and suppress the ions in each window [45]. An independent 10-nanosecond simulation was performed for each window. The potential of the mean force (PMF) curve was produced using the g_wham tool of Gromacs 5.0, which implements a weighted histogram [46].

## 3. Results and Discussion

### 3.1. Translocation Phenomenon 

The adsorption process of GQDs on different cell membranes was studied using MD simulations. The initial GQDs were placed 4.5 nm above the center of mass of the cell membrane. As shown in Figure 2a, the GQDs first reached the surface of the POPC cell membrane, and this process was completed in approximately 10 ns. Once the GQDs arrived at the surface of POPC, they rapidly permeated through the cell membrane. In the simulation of the DOPE cell membrane system, the GQDs were absorbed by the cell membrane within approximately 80 ns. Similar to the permeation of GQDs in POPC membranes, the GQDs were found to reach the surface of the membrane and rapidly penetrate it. However, unlike the POPC and DOPE lipid membranes, the GQDs were captured by the POPE membrane at 20 ns but remained on the upper surface of the membrane. Figure 2b shows the change in the number of phospholipid atoms adsorbed on the surface (d < 0.5 nm) of the GQDs. For the POPC cell membrane, the number of atomic contacts reached a maximum of 495 in 15 ns and then remained stable at 500. In this time, GQDs completely entered the POPC membrane. Similarly, GQDs completely entered the DOPE membrane at 110 ns, and then, the atomic contact number remained at 450. However, for the POPE lipid membrane, because the GQDs always remained on the surface of the cell membrane, only one side of the atom was attached to the lipid membrane, and the number of atoms in contact with the membrane was relatively low. In addition, some water molecules were also trapped on the surface of the GQDs and phospholipid molecular membrane head groups, which also led to low contact between the GQDs and POPE atoms.

To visualize the translocation process of GQDs on lipid membranes, instantaneous screenshots of GQDs on membranes at different time points are shown in Figure 3. For the POPC and DOPE membranes, permeation into the cell membrane by GQDs from the surface was similar. Specifically, the GQDs were inclined at an angle to the POPC and DOPE membrane surfaces at 14 and 96 ns, respectively. The GQDs were first inserted into phospholipid molecules and then rapidly entered the cell membrane. Meanwhile, they maintained an orientation parallel to the conformation of the phospholipid molecule and stably adhered to the interior of the cell membrane. Figure 4 shows the change in the angle between the GQDs and the lipid membrane over time. The initialized GQDs moved freely in the liquid phase water, with large changes in the angle. When entering the cell membrane, the GQDs were maintained at an angle of 80° relative to the membrane surface. For the POPE membrane, the angle between the GQDs and the membrane (Figure 4b) showed a significant change at 130 ns, and the GQD angle was close to 50°. When the simulation reached 172 ns, the angle decreased to 10°, and the adsorption on the surface of the cell membrane was maintained until the end of the simulation.

### 3.2. Effect of the Phospholipid Molecule Concentration on Translocation

The MD simulation showed that the GQDs could not enter the POPE cell membrane. To explain this phenomenon, the cell membranes were analyzed based on the total number of phospholipid molecules. Due to the different types of phospholipid molecules, the density of phospholipid molecules per unit area was different. With the size of the initial box kept the same, the density of phospholipid molecules in POPE was the highest among the three cell membranes, with one box containing 342 molecules (Table 1). GQDs entering the cell membrane were inserted between the phospholipid molecules in the membrane. The size of the gap between adjacent phospholipid molecules decreased with the increasing density of phospholipid molecules. Therefore, it is more difficult for GQDs to cross this energy barrier and enter the cell membrane with a high phospholipid membrane density.

To explore whether the change in phospholipid molecular density in the membrane would affect the penetration of GQDs, we constructed a low-density POPE membrane (POPE-LD). Without changing the number of phospholipid molecules (i.e., 342), the density of phospholipid molecules in the system can be changed by expanding the size of the outer frame of the cell membrane. The size of the simulated box was expanded from 9.0 × 9.0 × 10 nm^3^ to 9.6 × 9.6 × 10 nm^3^, and the position of GQDs was kept unchanged. By adding water molecules, the cell membrane was completely balanced for 10 ns under NVT conditions. During the balancing process, the energy and temperature of the system tend to be stable (see Appendix A for details). This shows that after 10 ns NVT simulation, the cell membrane tended to equilibrium.

The simulation results are shown in Figure 5. Similar to DOPE and POPC, the GQDs reached the surface of the membrane at 40 ns and had completely entered the membrane at 50 ns. Similarly, the fully stabilized GQDs remained in the upper center of the membrane. 

By expanding the size of the simulation box in the xy direction, the density of phospholipid molecules is reduced in the plane. In order to compare the density of phospholipid molecules more clearly, as shown in Figure 6, the density of the POPE membrane in a high-density state and a low-density state was calculated. There are large gaps between the phospholipid molecules with the lower density, which makes it easier for GQDs to enter the interior.

### 3.3. Free Energy Profile

To further explain the translocation mechanism of the GQDs on the cell membrane, PMF was used to calculate the shift in free energy of GQDs on POPE membranes, as shown in Figure 7. In the systems, the center of the cell membrane was defined as the zero point of free energy, and the surface free energy of GQDs from the liquid phase to the cell membrane decreased. It shows that GQDs are more inclined to adsorb on the surface of cell membranes than liquid phase water. However, compared with the POPE-LD membrane at low density, GQDs need to cross the higher energy barrier on the surface to enter the POPE membrane, which reduces the probability of GQDs penetrating the membrane surface of high-density phospholipid molecules. In addition, it is noted that the lowest points of free energy are located above the inner center of the cell membrane. In the PMF curve of POPC and DOPE cell membranes at low density (see Appendix A), the free energy change of GQDs entering the cell membrane decreases, and the surface energy barrier is low. The lowest point of free energy is located inside the cell membrane, which is consistent with our MD results.

GQDs can penetrate cell membranes, and interactions between them include electrostatic (Ele) and Van der Waals (VDW) forces. To explore the interactions between the GQDs and the cell membranes in all systems in the MD simulation, as shown in Figure 8, the changes in these two forces with time were calculated. Among them, VDW and Ele forces were zero, indicating that the GQDs were not yet in contact with the cell membrane. When touching the surface of the membrane, the GQDs were pulled into the membrane by the effects of VDW and Ele forces. From the overall calculation results, it can be observed that the effect of VDW force was significantly stronger than that of Ele force, which is the main driving force for GQDs to enter the cell membrane. However, in the high-density POPE system, the effects of VDW and Ele forces were weaker than those of the other systems, which is not sufficient to drive GQDs into the cell membrane.

## 4. Conclusions

MD simulations were used to explore the interactions between GQDs and different cell membranes, as well as to study the translocation process of GQDs in these settings. The simulation demonstrated that GQDs can easily permeate POPC and DOPE lipid membranes. For POPE lipid membranes, the GQDs can only be adsorbed onto the surface. This is consistent with the PMF calculation of the displacement of GQDs on this cell membrane. POPE has a high energy barrier, which makes it difficult for GQDs to cross. Analysis of the lipid membrane structure revealed that the density of phospholipid molecules is a key factor affecting the ability of GQDs to permeate the cell membrane. High-density POPE membranes are very difficult for GQDs to penetrate, whereas low-density POPE membranes allow GQDs to be transported into the intracellular space. The lipid density will have a great influence on the energy barrier affecting GQDs’ diffusion into the membrane. In real cell systems, the lipid densities are not uniform due to thermal fluctuation and cell movement. Some spots on the cell membrane will have a lower density, and these low-density spots will be the pathway for GQDs to diffuse into the cell membrane. This study provides a theoretical reference for further studies on the toxicity of GQDs and their transport across membranes. 

## Figures and Tables

**Figure 1 membranes-12-00753-f001:**
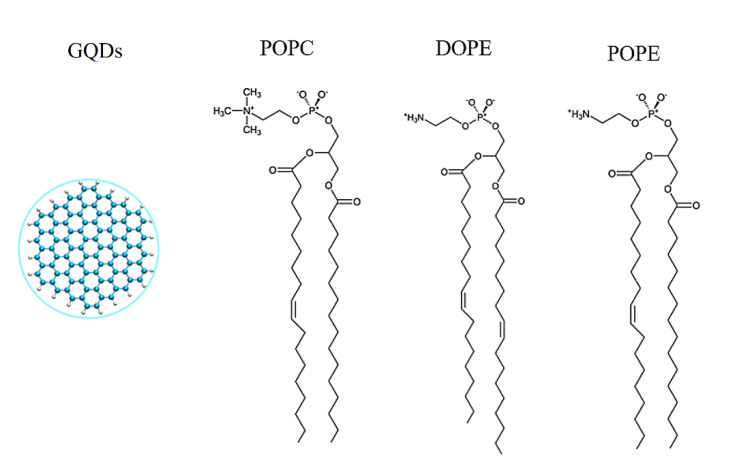
Structural details of three phospholipids and GQDs.

**Figure 2 membranes-12-00753-f002:**
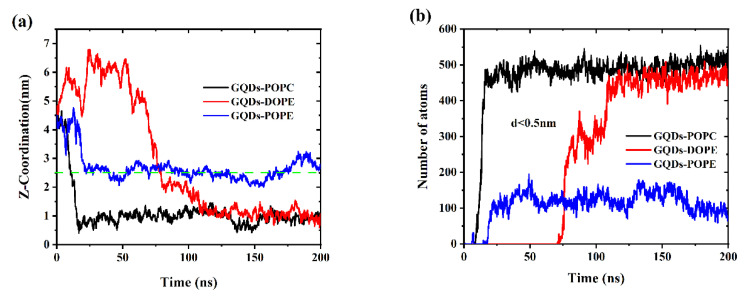
(**a**) Change in distance between the GQDs and the cell membrane in the z-direction. (**b**) Changes in contact numbers between GQDs and cell membrane atoms.

**Figure 3 membranes-12-00753-f003:**
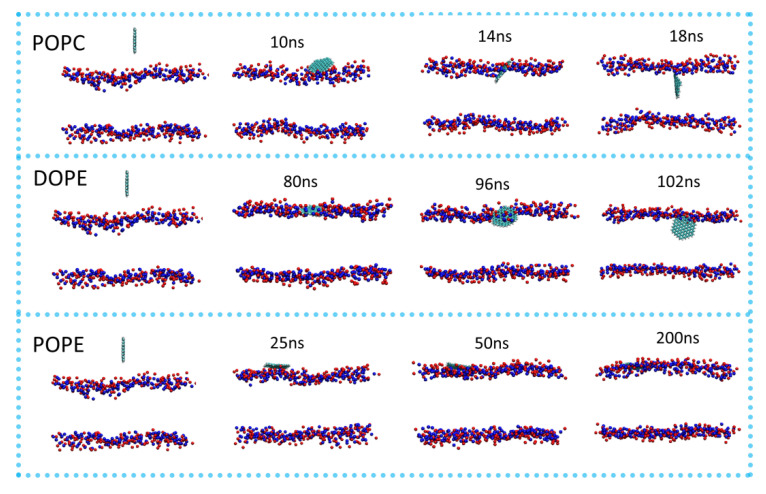
Permeation of GQDs through different cell membranes. P atoms in the cell membrane are shown as blue spheres, and N atoms are shown as red spheres. All instantaneous screenshots were produced using VMD software.

**Figure 4 membranes-12-00753-f004:**
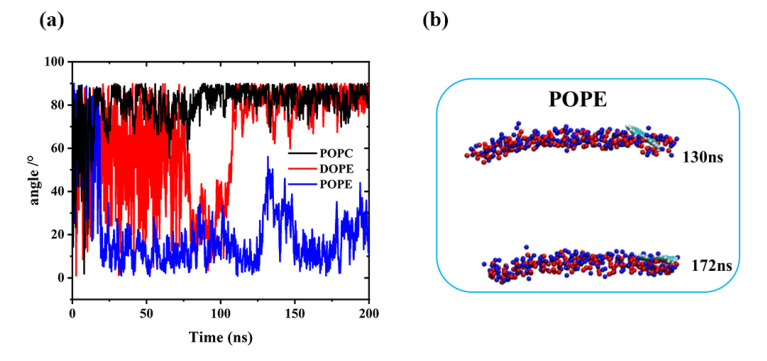
(**a**) Change in the plane angle between GQDs and the cell membrane during simulation. (**b**) Angle of GQDs on POPE cell membrane at 130 and 172 ns.

**Figure 5 membranes-12-00753-f005:**
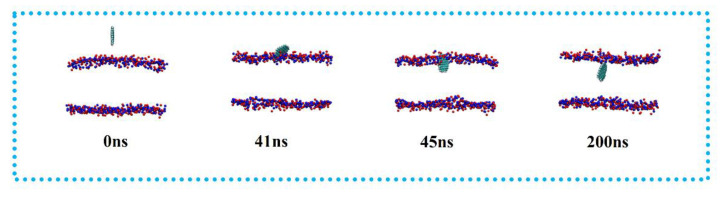
Penetration of GQDs on POPE-LD membrane.

**Figure 6 membranes-12-00753-f006:**
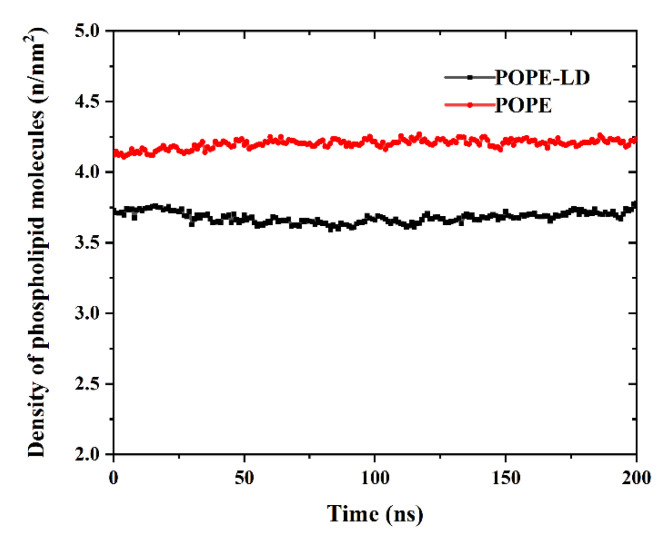
Change in phospholipid molecular density of POPE membrane.

**Figure 7 membranes-12-00753-f007:**
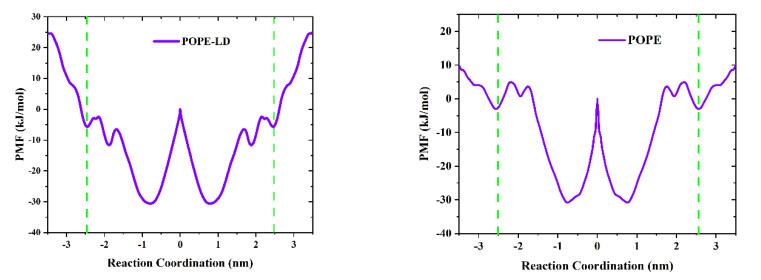
The average force (PMF) potential of GQD transport through the membrane. The green dotted line indicates the two boundaries of the cell membrane.

**Figure 8 membranes-12-00753-f008:**
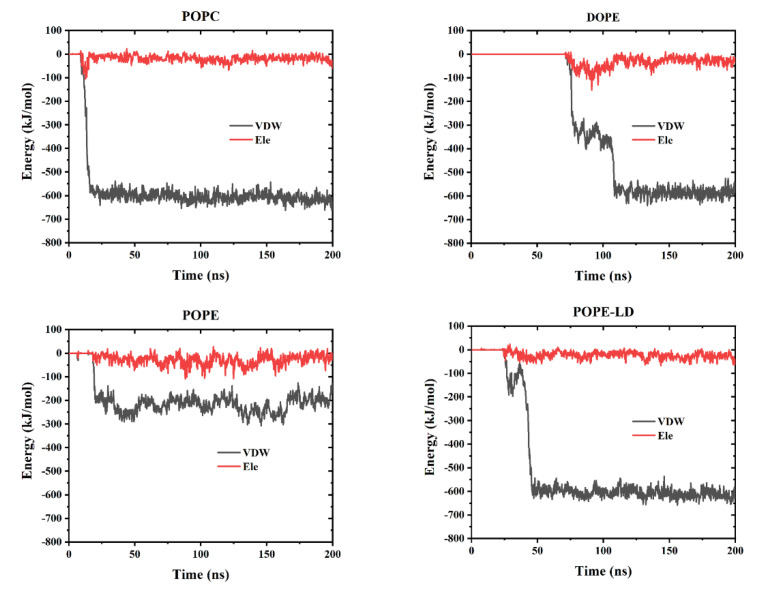
The interaction energy between GQDs and cell membranes includes van der Waals interaction (VDW) and classical interaction (Ele).

**Table 1 membranes-12-00753-t001:** Simulation details of the phospholipid molecules.

System	No. of Atoms	Molecular Formula	Molecular Number	$\mathrm{Surface}\;\mathrm{Area}\mathrm{Per}\;\mathrm{Lipid}\;({Å}^{2})$
POPC	134	$C_{42}H_{82}{\mathrm{NO}}_{8}P$	274	68.3
DOPE	129	$C_{41}H_{78}{\mathrm{NO}}_{8}P$	316	63.4
POPE	125	$C_{39}H_{76}{\mathrm{NO}}_{8}P$	342	58.8

## Data Availability

The data presented in this study are available on request from the corresponding author.

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
