# Peer review of "Molecular Dynamics Simulation of Transport Mechanism of Graphene Quantum Dots through Different Cell Membranes"

_membranes, 2022, doi:10.3390/membranes12080753_

Round 1

Reviewer 1 Report

The authors studied the transmembrane dynamics of a graphene quantum dot molecule through a model cell membrane (POPE, DOPE and POPC respectively) using atomistic molecular dynamics simulations. The equilibrium structure and PMF calculations are performed to evaluate the interplay between GQD and membrane. They compared the translocation mechanism of the three types of membrane and found that the POPE lipid membrane shows a higher energy barrier for GQD to translocate compared with POPE and DOPE due to the difference in their packing density. The overall design of the simulation systems is reasonable and the presentation and analysis of the simulation results are clear and easy to follow. The findings are also interesting regarding the packing density difference in POPE membrane could lead to a dramatic difference in the energy barrier of the GQD transmembrane process. However, the simulation method has a serious problem. I would not recommend this paper to be published on Membranes unless the authors addressing the following issues:

(1)    The simulation details and interaction parameters should be included in this paper (could be placed into SI) instead of referring to your previous paper [32-34? Or other], which is not exactly the same system setup as in this paper. Doing that will save a lot of time for people who read your paper.

(2)    In fig 1, the authors provided the chemical structure of the three lipid molecules, and at the end of the lipid tail they labeled C18, C16 to represent the ending carbon atom. This is a little bit misleading, and it looks like there is a long alkane chain attaching to the end of the lipid tail. Please remove these numbers to make it clear.

(3)    When performing a membrane simulation, it is very important to setup the system in a correct physical environment. Membrane is an anisotropic system where the lateral direction and normal direction has different equilibrium environment. Therefore, when equilibrium a membrane system, it is very important to set the pressure coupling to anisotropic or semi-isotropic where the xy direction and z direction are coupled separately. From the simulation results and the discussion about putting different number of POPE molecules into the system to study the density effect. I have a feeling that the pressure coupling method for equilibrating the membrane could be wrong in current simulation. Since the authors didn’t provide the simulation details that detailed enough, I guess the pressure coupling algorithm used in this work is isotropic. Otherwise, changing the number of lipids will not affect the packing density of the lipids. If that is the case, then that will be a serious problem, and SADLY, you have to recalculate your simulation and check if the conclusion of this paper still hold. I don’t know where you get the data of the surface area per lipids but the first thing for a membrane simulation is to check if the area per lipids matches the experimental results or not. Please add this section into this paper to validate whether the forcefields parameters and simulation conditions are set properly.

(4)    In fig 2b, the authors provided the contact number between the membrane and the GQD and the result for POPE case is much lower than the other two samples. Are there any water molecules being trapped between the GQD and membrane head groups?

(5)    Reference inconsistencies are found, such as ref 46 at page 4 section 2.2. The forcefield parameters cannot come from ref 46 which is related with umbrella sampling. Please check other references as well. The format of the reference list is messy, and the references are not in the same format. Some special characters (%) are also found and should be deleted and some typos about the author names (ref 42) are found. Please check them again to make sure they are in the required format in the template of Membranes.

(6)    A serious language polishing is needed.

Author Response

Thank you very much for your comments. We sincerely accept your suggestions and have fully modified the article according to your comments. Please see the attachment.

Reviewer 2 Report

The presented manuscript investigates the transport mechanism of graphene quantum dots through different cell membranes using molecular dynamics simulations. Through the free energy calculations, the translocation capability of GODs was evaluated, the authors found that the phospholipid molecules played important role for the entering process of GODs. I would recommend publication of this manuscript after the following points are carefully considered and some clarifications are provided.

1.     Normally a TOC figure doesn’t need a caption, however, the authors provided a figure caption for the TOC figure and put the description in the Abstract section, which reduced the significances and highlights of the study. The authors should re-consider the Abstract section.

2.    Section 2.1, “the structure of the GODs was optimized using Gaussian 03.” What is the level of theory? What is the basis set? Have you checked the frequency analysis for avoiding any improper structures (imaginary frequency)?

3.     Figure 5, if profile becomes asymptotic at |R| > 3.5, normalize profile to set asymptotic value to 0.

4.     In Fig. 8, I guess no charges on the surface of the GQD would lead to nearly no Ele interaction, is it correct? If so, is it true in reality (experiment)?

5.     Why is the translocation process in DOPE and POPE so different, from the structural point of view, these two structures are very similar (cf. Fig. 1). For example, the author attributed the translocation capability of GOD to the density of phospholipid molecules for POPE systems, so could the authors provide more direct evidence? Such as the distribution distance of phospholipid molecules (g(r) for the Phosphorus atoms)

Minor points:

1.    Page 2 Paragraph 2: “Many studies reported that GQDs have potential applications in medicine and can enter cells through direct infiltration, however, the substantial variation between organisms in terms of cell membrane structure should be considered.” Citations were missing.

2.     The number 120802 in Table 1 was missing a comma.

3.  Page 5 Paragraph 1: “Fig. 1b shows the change in the number of phospholipid atoms adsorbed on the surface (d < 0.5 nm) of the GQDs with respect to simulation time.” Should be Fig. 2b.

4.     The green dashed line on Fig. 2a has to be explained in the caption.

5.   Page 8 Paragraph 2: “To explore the interactions between the GQDs and the cell membrane in the five systems in the MD simulation”, should be four systems.

6.     Check the format of Ref 6 and 42, the “%” symbols are presented.

Author Response

(The authors gave the same response as above.)

Round 2

Reviewer 1 Report

OK, let’s go over all the issues I found (with solution as bonus) and why I suggest you recalculate your simulations:

(1)    If I understand correctly, the self-assembling process should start from equilibrium state, i.e., the membrane should be in equilibrium state before the GQD begin to move. From your descriptions (in page 7) for preparing the POPE-314 sample, you mentioned that after extracting (or removing) 28 lipids from POPE-342, you didn’t relax the simulation system but directly let the GQD diffuse while equilibrium the simulation system. Therefore within 100 ns, although the density increased, the GQD has already diffused into the system (the POPE-314 is not fully equilibrated until 200 ns, shown from fig 7). In other words, the self-assembly of GQD is carried out under non-equilibrium state, and you allow the membrane and GQD equilibrating together. It is unreasonable/meaningless to perform a simulation like that. Could you find a physical/experimental procedure to perform experiments like that?   

(2)    Let’s talk about the difference in the lipid density between sample POPE-314 and POPE-342. the density difference between POPE-314 and POPE-342 could mean that your cell membrane needs more time to equilibrate. Under the correct ensemble condition (anisotropic NPT), the lipid density should be the same for the two samples. You need to extend your simulation much longer to check this. If they are not in the same density, it means more time needed to relax. I can understand the membrane simulation is slow, yes, I cannot agree more, but we need to make sure we do it properly right? You cannot draw any conclusion from a not-equilibrated-yet system.

(3)    If you still cannot get the density match between these two samples, yes, it could happen due to the statistical error. But the density cannot differ too much. If these two samples have similar density, I am sure you will get similar PMF curve instead of the one you get in figure 8.

(4)    Regarding the PMF curve you get from WHAM calculation from umbrella sampling. What is the ensemble condition you applied when doing the umbrella sampling? (NVT or NPT). I guess they are done in NVT ensemble since NPT ensemble will impose additional fluctuations to the spring and alter the statistical accuracy of COM location. If NVT ensemble is the way you perform the umbrella sampling, then the dynamic evolution of POPE-314 is meaningless since they are not in the same ensemble conditions (NVT vs NPT). The conclusion will be completely different. Therefore, logically the MD simulation and PMF comparison between POPE-314 and POPE-342 is meaningless because they are different systems.

(5)    OK, let’s discuss what is the correct way to do the simulations. I can understand your conclusion and the story you want to tell. The lipid density will have great influence on the energy barrier of GQD diffuse into the membrane. In real cell system, the lipid densities are not uniform due to the thermal fluctuation and cell movement. Some spots on the cell membrane will have lower density, these low-density spots will be the pathway for GQDs diffuse into the cell membrane. When preparing the low-density spot in simulation, the correct way is to deform the simulation box in xy direction or remove the lipids (from 342 to 314) and equilibrate the system under NVT (not anisotropic NPT you mentioned in the text) ensemble condition. The lipid density will decrease as expected. But still make sure to fully equilibrate the system under NVT condition and monitor the energy as needed. Then perform the GQD trans-membrane dynamics and PMF calculations. Doing that will make it logically consistent and matching the real physical process.

Author Response

 Thank you very much for your comments. We sincerely accept your suggestions and have fully modified the article according to your comments.Please see the attachment.

Round 3

Reviewer 1 Report

The PMF curve for POPE is now reasonable and the reason for GQD staying at the membrane surface is also explained properly. However, compared with previous version, why the authors removed PMF curves for other two types of lipids. I suggest authors put the PMF curves side by side to compare the structural differences induced free enegy changes. The formatting (figures, texts) for this version also needs a deep polishing (alignment issue). Also, please read through the manuscript to make sure all descriptions are consistent. In fig 5, the snapshsots are refered to the previous simulation results, please update to the correct one. Language polishig is also a must. Once these corrections are made, the manuscript could be accepted.

Author Response

Thank you very much for your valuable comments. Please see the attachment.
